# Fine Mapping and Identification of *BnaC06.FtsH1*, a Lethal Gene That Regulates the PSII Repair Cycle in *Brassica napus*

**DOI:** 10.3390/ijms22042087

**Published:** 2021-02-19

**Authors:** Kai Xu, Yujin Wu, Jurong Song, Kaining Hu, Zengxiang Wu, Jing Wen, Bin Yi, Chaozhi Ma, Jinxiong Shen, Tingdong Fu, Jinxing Tu

**Affiliations:** National Center of Rapeseed Improvement, National Key Laboratory of Crop Genetic Improvement, Huazhong Agricultural University, Wuhan 430070, China; xk@webmail.hzau.edu.cn (K.X.); wuyujin@webmail.hzau.edu.cn (Y.W.); sjr2015@webmail.hzau.edu.cn (J.S.); hukaining@gmail.com (K.H.); wuzengx@webmail.hzau.edu.cn (Z.W.); wenjing@mail.hzau.edu.cn (J.W.); yibin@mail.hzau.edu.cn (B.Y.); yuanbeauty@mail.hzau.edu.cn (C.M.); jxshen@mail.hzau.edu.cn (J.S.); futing@mail.hzau.edu.cn (T.F.)

**Keywords:** lethal mutant, map-based cloning, FtsH1 hydrolase, PSII repair cycle

## Abstract

Photosystem II (PSII) is an important component of the chloroplast. The PSII repair cycle is crucial for the relief of photoinhibition and may be advantageous when improving stress resistance and photosynthetic efficiency. Lethal genes are widely used in the efficiency detection and method improvement of gene editing. In the present study, we identified the naturally occurring lethal mutant 7-521Y with etiolated cotyledons in *Brassica napus*, controlled by double-recessive genes (named *cyd1* and *cyd2*). By combining whole-genome resequencing and map-based cloning, *CYD1* was fine-mapped to a 29 kb genomic region using 15,167 etiolated individuals. Through cosegregation analysis and functional verification of the transgene, *BnaC06.FtsH1* was determined to be the target gene; it encodes an filamentation temperature sensitive protein H 1 (FtsH1) hydrolase that degrades damaged PSII D1 in *Arabidopsis thaliana*. The expression of *BnaC06.FtsH1* was high in the cotyledons, leaves, and flowers of *B. napus*, and localized in the chloroplasts. In addition, the expression of *EngA* (upstream regulation gene of FtsH) increased and D1 decreased in 7-521Y. Double mutants of *FtsH1* and *FtsH5* were lethal in *A. thaliana*. Through phylogenetic analysis, the loss of *FtsH5* was identified in *Brassica*, and the remaining *FtsH1* was required for PSII repair cycle. *CYD2* may be a homologous gene of *FtsH1* on chromosome A07 of *B. napus*. Our study provides new insights into lethal mutants, the findings may help improve the efficiency of the PSII repair cycle and biomass accumulation in oilseed rape.

## 1. Introduction

The discovery and utilization of lethal genes is of great significance to the progress of molecular breeding. Lethal genes are widely used to detect the efficiency of gene editing systems in plant, which the editing efficiency can be observed easily at the seedling stage. As a powerful tool for genome editing, the CRISPR-Cas9 system was first successfully applied to plants by editing an albino gene *OsPDS* in rice, which exhibited an expected phenotype in T_0_ generation [1]. Using the *zb7* gene as an editing target, a large number of lethal seedlings observed in T_0_ generation, which proved that the dmc1 promoter-controlled (DPC) CRISPR/Cas9 is a high-efficiency editing system in maize [2]. Similar studies in *Arabidopsis thaliana* (*A. thaliana*) [3], *Triticum aestivum* [4], *Gossypium* spp. [5,6], *Nicotiana tabacum* [7], and other crops are often reported. However, few studies about gene cloning for lethality were reported for rapeseed due to the complex genome of *Brassica napus* (*B. napus*) and the limited reference genome before the release of *B. napus*, “ZS11”. Therefore, the functional analysis of lethal mutants in rapeseed has potential value for improving gene editing efficiency and advancing technical innovation in molecular breeding.

Approximately 1 to 1.5 billion years ago, chloroplasts evolved via the endosymbiosis of an ancient relative of an extant cyanobacterium, and they have become an indivisible part of the photosynthetic apparatus of terrestrial plants and algae [8,9]. The chloroplasts are largely responsible for photosynthesis, carbon metabolism, and fatty acid synthesis. However, the chloroplast genome is very small, and 95% of chloroplast proteins are encoded by nuclear genes [10]. The coordination of nuclear genes and chloroplast genes maintain the normal functions of chloroplasts, and the isolation and research of nuclear genes will contribute to further understanding the chloroplast developmental mechanism. Chloroplast-related mutants are excellent genetic resources to study the mechanisms of photosynthesis, chlorophyll biosynthesis, chloroplast structure, function, and development.

Photosystem II (PSII) is one of the most vulnerable parts of the plant photosynthesis system. Light energy is absorbed in the reaction center of the PSII complex, but it concurrently causes photodamage. A major target of this photodamage is the D1 protein of PSII, but the stability of PSII is necessary for the appropriate development of plants [11]. Thus, the PSII repair cycle is formed in photosynthetic organisms, and it involves the following processes: the partial decomposition of the PSII complex, selective degradation of photodamaged D1, synthesis of new D1, and reassembly of the functional PSII complex. The entire process is significant for the recovery of photosynthetic efficiency [12]. In seed plants, Deg and filamentation temperature sensitive protein H (FtsH) proteases together complete the degradation of damaged D1 [13,14,15,16]. The *A. thaliana FtsH* gene family comprises 12 members (*FtsH 1–12*) and five FtsHi members which have lost the function for protein hydrolysis (*FtsHi 1–5*) [17]. However, FtsH members are the major thylakoid membrane proteases. Four FtsH proteins form a heterohexameric complex in the thylakoid membrane, comprising type A (includes FtsH5 and FtsH1) and type B (includes FtsH8 and FtsH2) [18]. At least one subunit of each type is required for chloroplast development and normal photosynthesis, and double mutants that result in the loss of either type A or type B proteases are lethal [19,20]. The appropriate turnover of FtsH is important for the maintenance of functional PSII repair. Overexpression of *EngA* (EngA-OX) results in the variegation of leaves and the accumulation of more cleaved D1 fragments. *EngA* may negatively regulate FtsH stability by interacting with its ATPase domain [21].

In this study, we identified a new lethal mutant 7-521Y with yellow cotyledon in *B. napus*. We fine-mapped and functional confirmed the candidate gene *CYD1*, which encodes an FtsH1 protease involved in the PSII repair cycle. Through phylogenetic analysis, we identified the loss of *FtsH5* in *Brassica*; the remaining *FtsH1* was required for plant survival. This study has important guiding significance for genotype selection of oilseed rape.

## 2. Results

### 2.1. Characterization of the 7-521Y Mutant

The natural mutant 7-521Y was found in the breeding material of the 7-521 line. The mutant had a yellow cotyledon, could not grow true leaves, and died after 7–14 d of growth (Figure 1a,b). The chlorophyll a, chlorophyll b, and carotenoid (Car) contents of the etiolated seedlings were significantly lower than those of normal 7-521G individuals (Figure 1c). No significant difference was detected in the chlorophyll a/b ratio between 7-521G (3.920 ± 0.054, mg/mg) and 7-521Y (4.468 ± 0.465, mg/mg), but the carotenoid/chlorophyll ratio of 7-521Y (0.592 ± 0.068, mg/mg) was significantly higher than that of 7-521G (0.186 ± 0.000, mg/mg) (Appendix A). The chloroplast structures of 7-521G and 7-521Y were analyzed by transmission electron microscopy (TEM) (Figure 1d). The chloroplasts of 7-521G developed large starch grains; however, the chloroplasts of 7-521Y did not develop any starch grain. Plump chloroplast structures and stacked thylakoid membranes were observed in 7-521G, but were not observed in 7-521Y (Figure 1d and Appendix A). 

Among the 484 progenies of 7-521, there were 123 etiolated seedlings and 361 green seedlings, with the genetic segregation ratio of 3:1 (χ^2^ = 0.025 < χ^2^_0.05,1_ = 3.84). To verify that this trait of etiolated cotyledons was controlled by one or two genes, the heterozygous 7-521G was utilized as the paternal line in a cross with the Bing 409 line, and then the segregation of eight F_2_ populations using seedling color was analyzed. Five out of eight F_2_ populations showed a Mendelian segregation ratio of 15:1 (Appendix A), which fits with a two-gene model. The controlling genes were named *CYD1* and *CYD2*. Double-recessive genes controlled this lethal trait, 7-521 carried heterozygous alleles for *CYD1* and homozygous recessive alleles for *CYD2*. 

### 2.2. Whole-Genome Resequencing and Gene Mapping for the Lethal Gene CYD1

Separately mixed normal and yellow plants in the 7-521 line were used for whole-genome resequencing, for which the average sequencing coverage depths were 52.67× and 56.22× in the 7-521G and 7-521Y pools, respectively. Resequencing generated approximately 93.5 Gb of clean data after filtering 102.3 Gb of raw data that consisted of 303,754,772 reads from 7-521G and 324,204,540 reads from 7-521Y. Based on the ∆SNP-index value, 4,177,432 SNPs were called, and a Manhattan map (Figure 2a) was used to illustrate each chromosome. Three potential regions were identified: chromosome A03 (6.85–6.90 Mb), chromosome A09 (22.78–22.90 Mb), and chromosome C06 (20.48–36.40 Mb). Indel markers were designed to screen 7-521G and 7-521Y DNA pools to identify polymorphic markers. Only markers on chromosome C06 showed polymorphisms between the two DNA pools. All results indicated that the major candidate gene *CYD1* of 7-521 may be on chromosome C06 of *B. napus* (Figure 2b).

To reveal the molecular mechanisms of the etiolated-cotyledon phenotype, we isolated the key gene *CYD1* using map-based cloning, and SSR markers were designed. We selected 8777 individuals with etiolated cotyledons to map *CYD1*. *CYD1* was initially located on chromosome C06, an interval between markers S361 and S342 (Figure 3a), and corresponding to a 428 kb region of the *B. napus* ZS11 physical map. Using a larger population of 15,167 etiolated seedlings, *CYD1* was then fine-mapped to a 29 kb genomic region between SSR markers S481 and S342 (Figure 3b). The candidate interval contained five annotated open reading frames in the ZS11 reference genome (Figure 3c). Annotation of these genes (Table 1) indicated that *BnaC06G0341000ZS* was highly orthologous to *AT1G50250* in *A. thaliana*. *AT1G50250* encodes FtsH1 that regulates the PSII repair cycle, and *FtsH1*/*FtsH5* double mutants had the etiolated-cotyledon phenotype.

A comparison of *BnaC06.**FtsH1* between 7-521G and 7-521Y revealed that *BnaC06.**FtsH1* was missing in etiolated individuals. To confirm if the deletion cosegregated with etiolated cotyledons, SCAR markers FtsH1-C6GFL-F and R, were developed on the full-length genomic DNA (gDNA) of *BnaC06.**FtsH1* (Appendix A). This marker completely cosegregated with the etiolated cotyledons of 15,167 individuals. Therefore, it can be speculated that *BnaC06.FtsH1* is a candidate gene of *CYD1*. 

### 2.3. Functional Confirmation by B. napus Transformation

To confirm whether the deletion of *BnaC06.FtsH1* was the causal mutation, the allele *BnaC06.FtsH1* of 7-521G driven by native promoter was cloned and introduced into the calli of the etiolated seedlings using *A. tumefaciens*-mediated transformation. Six positive seedlings were obtained by PCR identification; these positive seedlings were similar to 7-521G in their normal growth, development, and reproduction (Figure 3d). Phenotypic segregation occurred in the transgenic T_1_ generation, and the PCR results showed that all the recovered phenotypes were positive seedlings and etiolated individuals were negative seedlings (Figure 3e). These results suggest that the deletion of *BnaC06.FtsH1* was responsible for the etiolated-cotyledons observed in 7-521Y.

### 2.4. Subcellular Location and Expression of BnaC06.FtsH1

The expression of *BnaC06.FtsH1* was detected by qRT-PCR. According to the results, the transcription level of *BnaC06.FtsH1* was low in the roots, hypocotyls, stems, flower buds, and siliques, but high in the cotyledons, leaves, and flowers (Figure 4a). Driven by the *BnaC06.FtsH1* native promoter, the GUS reporter gene was expressed in *A. thaliana* to monitor the expression of *BnaC06.FtsH1*. Multiple transgenic plants were obtained, and the GUS staining results showed that *BnaC06.FtsH1* was expressed in almost all tissues including cotyledons, young leaves, and flowers, and this was almost consistent with the qRT-PCR results (Figure 4b–f). 

To explore the subcellular localization of BnaC06.FtsH1, GFP was fused to the C-terminus driven by the 35S promoter and transiently expressed in Arabidopsis mesophyll protoplasts. As expected, the BnaC06.FtsH1 signal closely overlapped with the chlorophyll autofluorescence signal, illustrating that BnaC06.FtsH1 was localized in the chloroplasts (Figure 4g).

### 2.5. Deletion of CYD1 Influences the Expression of EngA and PsbA in B. napus 

Through sequence alignment, we identified four homologs of *EngA* (*AT3G12080*) in *B. napus* ZS11: two in the A genome (*BnaA05G0428200ZS* and *BnaA01G0369900ZS*) and two in the C genome (*BnaC05G0481000ZS* and *BnaC01G0461600ZS*). The structural diagram for the four genes was mapped (Figure 5a). The gene sequences of *BnaA01G0369900ZS* and *BnaC01G0461600ZS* were too short, and their mRNA expression could be hardly detected in 7-521G or 7-521Y. However, the expression of homologs in A05 and C05 was higher in the 7-521Y individuals than in the 7-521G individuals (Figure 5b). There are two homologs of *PsbA* (the coding gene of D1) in ZS11: *BnaA01G0425500ZS* and *BnaA06G0201300ZS*. The CDS sequences of the two copies were identical, and we could not design-specific primers to distinguish them. However, the total expression of *Bna.PsbA* was relatively decreased in the 7-521Y individuals (Figure 5b). This indicates that the deletion of *CYD1* increased the expression of *EngA* and decreased the expression of *PsbA*, and that the three genes may interact in *B. napus*.

### 2.6. CYD2 May Be the Homologous Gene of FtsH1 on Chromosome A07

A previous study has indicated that double mutants with complete losses of either type B (*FtsH8* and *FtsH2*) or type A (*FtsH5* and *FtsH1*) were lethal [20]. The full-length protein sequences of these four genes and their eigth closest homologs were extracted for further phylogenetic analysis (Figure 6a). These results showed that *FtsH2* and *FtsH8* had 2 and 4 homologous genes in ZS11, respectively. There were six *FtsH* homologous copies of the type B proteins to help maintain the stability of the PSII repair cycle in *B. napus.* However, there was no homologous gene of *FtsH5* in ZS11, and only two copies of *FtsH1* on chromosome C06 and chromosome A07 determine the normal function of type A proteins. We speculated that the 7-521Y mutant was caused by the abnormal function of type A proteins. SSR markers were designed on the homologous segment of chromosome A07 and multiple polymorphic markers were found (Appendix A). We preliminarily concluded that *CYD2* is located on chromosome A07 and may be the homologous gene of *FtsH1*.

### 2.7. Phylogenetic Analysis of FtsH1 and FtsH5 in Brassica

To further illustrate the impacts of the loss of *FtsH5*, the full-length protein sequences of type A (*FtsH1* and *FtsH5*) and their closest homologs in *Brassica rape* (*B. rape*), *Brassica nigra* (*B. nigra*), *Brassica oleracea* (*B. oleracea*), *Brassica juncea* (*B. juncea*), and *B. napus* were used for phylogenetic analysis (Figure 6b). The results indicated that there was only one homologous gene of *FtsH1* and *FtsH5* in *Brassica* diploids (*B. rape*, *B. nigra*, and *B. oleracea*), and two homologs in *Brassica* allotetraploids (*B. juncea* and *B. napus*). These seven genes all had greater sequence similarity to *FtsH1*. *Brassica* diploids had only one *FtsH1* homologous gene of type A for the formation of a heterohexameric complex and a PSII repair cycle. 

## 3. Discussion

In this study, we characterized a natural mutant with etiolated cotyledons named 7-521Y, which grew abnormally and could not survive long term. Instead of conventional strategies for gene location using backcross (BC) and F_2_ populations, we directly used heterozygous selfing lines of 7-521 to locate *CYD1*. The mapping population often determines the frequency of exchange at a certain site [22]. Restricted by mapping population possibly, we used a large group of 15,167 lethal seedlings to fine-map *CYD1* to a 29 kb interval. Because 7-521y could not grow normally, it was impossible to obtain homozygous lethal mutants; therefore, we adjusted the traditional tissue culture method [23]. We planted a large number of heterozygous lines in M_0_, which were cultured in dark conditions for 5 d for the elongation of hypocotyls, and then transferred to light condition for 1 d to distinguish the mutant phenotype from the normal phenotype. Etiolated seedlings were cut for complementary experiments, and *BnaC06.FtsH1* was determined as the candidate gene. The results of this study will be significant for future research on lethal mutants.

FtsH plays a crucial role in the maintenance of thylakoid membranes [24]. FtsH1 participates in the formation of the hetero-hexameric complex and regulation of thylakoid membrane biogenesis [25]. Meanwhile, thylakoid membranes are densely packed with pigment–protein complexes of PSII and PSI [26], lack of chlorophyll and carotenoids in the mutant may cause PSII and chloroplast abnormalities. Therefore, plump chloroplast structures and stacked thylakoid membranes were not observed in mutant 7-521Y. The ratio of Car/Chl is frequently applied to assess stress tolerance of photosynthesizing tissues [27], shift of Car/Chl ratio in mutant may be an indicator of functional PSII dissasembly.

In photosynthetic organisms, the thylakoid FtsH complexes participate in the process of PSII repair cycle, particularly the degradation of damaged D1 [12,28,29]. *CYD1* may encode an FtsH1 hydrolase that is responsible for the degradation of D1. In the 7-521 strain, the deletion of *CYD1* in the mutant prevents the degradation of the damaged D1, and new D1 structural proteins cannot be synthesized, which blocks the PSII repair cycle for photosynthesis and causes the death of the mutant. However, the FtsH1 hydrolase only removes the light-damaged D1 protein and does not bind to the normal D1 protein [30]. We could not verify the interactions of *BnaC06.FtsH1* and D1 using the yeast two-hybrid assays. However, we found that the loss of *CYD1* increased the expression of *BnaA05.EngA* and *BnaC05.EngA* and decreased the expression of D1. *EngA* may negatively regulate the stability of FtsH1 and affect the degradation of damaged D1 by FtsH1 in *B. napus* [21].

Six *Brassica* species constitute the recognized “triangle of U” model [31], including three diploid species—*B. rapa* (A genome), *B. nigra* (B genome), and *B. oleracea* (C genome)—and three amphidiploid species resulting from pairwise hybridization—*B. napus* (AC genome), *B. juncea* (AB genome), and *Brassica carinata* (BC genome). *B. rapa* and *B. oleracea* originated from a common ancestor, and their genomes are very similar [32]. Therefore, the recessive traits controlled by two genes are often controlled by two duplicate genes in *B. napus* [31]. We speculated that *CYD2* may be a duplicated gene of *CYD1* on chromosome A07. Through marker development, it was confirmed that *CYD2* was located on chromosome A07 where *BnaA07.FtsH1* is located. *CYD2* may be *BnaA07.FtsH1*, but further verification is required. According to our results, there was a homologous gene of *FtsH1* in *B. rapa* and *B. oleracea,* respectively. *BnaC06.FtsH1* may have evolved from *Bol026111* (the orthologs of *FtsH1* in *B. oleracea*), while *BnaA07.FtsH1* may have evolved from *Bra004247* (the orthologs of *FtsH1* in *B. rapa*). 

In *A. thaliana*, *FtsH1* and *FtsH5* together determined the normal function of type A FtsH hydrolase, and we found that it was essential for at least one subunit of the PSII repair cycle [20]. Through phylogenetic analysis, we identified the loss of *FtsH5* in *Brassica*. In *B. juncea* and *B. napus*, the composition of type A proteins changed into two *FtsH1* homologous genes to ensure the normal degradation of the damaged D1. However, in *B. rapa*, *B. nigra*, and *B. oleracea*, type A consisted of only one *FtsH1* homologous gene, which was extremely detrimental to the survival of the plants. Any form of mutation in the *FtsH1* homologous genes may cause a lethal mutation phenotype similar to 7-521Y in *Brassica* diploid. This indicates that this gene is essential for the PSII repair cycle and the survival and normal growth of *Brassica*.

Although details of PSII repair cycle are still controversial, preventing photoinhibition is considered as a means to improve stress resistance and photosynthetic efficiency. Coordinating the degradation of D1 is considered an effective way to alleviate photoinhibition [33]. According to a recent study, the overexpression of *SlWHY1* enhances the synthesis of D1 in PSII and increases the chilling resistance of tomato [34], and increasing the expression of D1 via the nucleus can significantly increase the photosynthetic efficiency of *A. thaliana*, tobacco and yield of rice [35]. The deletion of *CYD1* decreased the expression of D1 in this study. It may be possible to adjust the expression of *CYD1* to accelerate the degradation of damaged D1 to improve the efficiency of the PSII repair cycle, thereby promoting photosynthesis and biomass accumulation in *B. napus*. In this study, we cloned *BnaC06.FtsH1* and identified the loss of FtsH5 in *B. napus*. We then speculated that the lethal phenotype of 7-521Y was caused by *BnaC06.FtsH1* and *BnaA07.FtsH1*. Because of the adjustment in our experimental methods, our research has reference significance for research on other lethal mutants. The mechanism analysis of 7-521Y also has potential value for the innovation of molecular breeding technology, especially gene editing in *B. napus*.

## 4. Materials and Methods

### 4.1. Genetic Analysis and Related Plant Materials

The mutant 7-521Y was found in the breeding material line 7-521 and mutated naturally. Green plants with heterozygous genotypes were selfed to obtain families, which were then used as segregating populations for mapping *CYD1* (Appendix A). The heterozygous genotype plants were then crossed with Bing409 to obtain F_1_ plants, which were self-crossed to obtain F_2_ segregation for analyzing heredity patterns. Plant descendants segregated at the chromosome A07 locus but with a recessive homozygous allele on the chromosome C06 locus were used to obtain F_3_-1 populations, which were used for fine mapping *CYD2*. All *B. napus* materials were planted in the experimental base at Huazhong Agricultural University in Wuhan. The planting density between rows was 20 cm and the row spacing was 30 cm.

### 4.2. Pigment Assay

Fresh cotyledons (approximately 0.15 g) were cut into pieces and placed into 10 mL of the extraction solution (96% ethanol: acetone = 1:8 (*v*/*v*)), and then placed in the dark for 48 h. The pigment content was determined spectrophotometrically (Mapada, Shanghai, China), quartz cuvettes with 10mm optical pathlength were used. The absorbance of chlorophyll solution was determined at 470, 645, and 663 nm, with the extraction solution as the control [36]. Three replicates were assessed, and the results were calculated using the following equations [30]:
Chl a = (12.21A_663_ − 2.81A_645_)/(50 × weight), Chl b = (20.13A_645_ − 5.03A_663_)/(50 × weight)
Car = (1000A_470_ − 3.27 (12.21A_663_ − 2.81A_645_) − 104 (20.13A_645_ − 5.03A_663_))/(11450 × weight)
where Chl a, Chl b, and Car represent the contents of chlorophyll a, chlorophyll b, and Carotenoid in leaves (mg/g), respectively. while A_663_, A_645_, and A_470_ represent the absorbance at 663, 645, and 470 nm, respectively. Weight is the weight of cotyledons (g).

### 4.3. Transmission Electron Microscope (TEM) Analysis

Fresh cotyledons of 7-521G and mutant plants 7-521Y were cut and fixed in a solution of 0.1 M phosphate buffer of pH of 7.4% and 2.5% glutaraldehyde (*w*/*v*); TEM analysis was then performed by H-7650 (Hitachi, Tokyo, Japan). The experiment was performed as described previously [37].

### 4.4. Whole-Genome Resequencing and Gene Mapping

Genomic DNA from 7-521G and 7-521Y bulks was subjected to whole-genome resequencing. The total DNA was extracted from cotyledons using the Plant Genomic DNA Kit (Tiangen, Beijing, China). To ensure the concentration and quality of gDNA, assessments were performed by Invitrogen Qubit 2.0 (Thermo Fisher Scientific, Waltham, MA, USA). Two DNA pools were used for the construction of libraries using TruSeq DNA Sample Preparation Kits (Illumina, San Diego, CA, USA). Then, a HiSeq 2000 system (Illumina, San Diego, CA, USA) was used to produce paired-end reads. Quality analysis and filtering of read datasets were performed using FastQC (http://www.bioinformatics.babraham.ac.uk/projects/fastqc/, last accessed date 7 December 2020) and Trimmomatic software [38], respectively. While IGV [39] was used as the genome browser. SNP and INDEL variant calling was performed using a previously described method [40], with the reference genome of *B. napus* “Darmor-bzh” [32]. The Δ (SNP index) was calculated by subtracting the SNP index of 7-521Y from that of 7-521G, and a Manhattan map was drawn using the ggpplot2 package in R [41]. The BSA method was applied to screen molecular markers, and the SSR markers were designed using an online tool (https://bioinfo.inf.ufg.br/websat/, last accessed date 7 December 2020) and synthesized (Tsingke, Wuhan, China), utilizing the reference genome sequence of *B. napus* “ZS11” [42]. We used 15,167 etiolated plants to map *CYD1*, in which the genomic DNA was extracted using the CTAB (hexadecyltrimethylammonium bromide) protocol [43]. The candidate interval sequences were submitted to the BnPIR website [42] and the Brassica Database (BRAD) [44] for BLAST analysis. The putative functions of candidate genes were adopted from their orthologs in *A. thaliana*, using the BLAST tool of the TAIR website (https://www.arabidopsis.org/, last accessed date 7 December 2020).

### 4.5. Plasmid Construction and Transformation for Complementation

To validate the function of *BnaC6.FtsH1*, a 6352-bp DNA fragment of the gene including the 2784 bp upstream sequence, 2357 bp coding region, and 1211 bp downstream sequence of *BnaC6.FTSH1* was amplified from 7-521G (primers FtsH1-CZQC-3F and FtsH1-CZQC-3R) and then cloned into the pCAMBIA2300 vector [45] using single restriction endonucleases (Pst I) and a ClonExpress II One Step Cloning Kit (Vazyme, Nanjing, China). We confirmed that the constructed vectors were correct by restriction digestion analysis and sequencing. Finally, the vector plasmid was transferred into GV3101 (*Agrobacterium tumefaciens* strain) by the electroporation method, using Electroporator 2510 (Eppendorf, Hamburg, Germany) in 100 Ω, 50 μF and 1800 V, and subsequently transferred into yellow individuals of the segregation lines. The hypocotyl of the mutants could be elongated in the dark. Tissue culture was performed as described previously [23]. We seeded the segregation population on M_0_ solid medium and cultured in the dark for 5–6 d, and then cultured under light condition for 1 d. The yellow plants were selected for complementary transformation. Kalamycin was found to inhibit callus differentiation and plant regeneration in our materials; therefore, it was not used for selection during the tissue culture process. Positive seedlings were confirmed by PCR (marker M13-47F and A3-YXJC-11R), and the genetic background of the positive seedlings was found to be homozygous recessive using markers S445, S474, S342, and S349.

### 4.6. RNA Extraction and Quantitative Real Time PCR (qRT-PCR)

Different tissues of Westar were sampled for total RNA extraction, using the RNAprep Pure Plant Kit (Tiangen, Beijing, China), including the hypocotyls, roots, stems, cotyledons, leaves, buds, siliques, flowers of Westar, and cotyledons of 7-521G and 7-521Y. Three biological replicates were used for all samples. Reverse transcription was conducted using the RevertAid First Strand cDNA Synthesis Kit (Thermo Fisher, Massachusetts, USA). Real-time PCR was performed in triplicate using the SYBR Green Realtime PCR Master Mix (Toyobo, Tokyo, Japan) in a CFX96 Touch Real-Time PCR (Bio-Rad, California, USA). Quantitative RT-PCR measurements were obtained using the relative quantification 2^−ΔΔCt^ method. Data are expressed as the mean of three biological replicates ±SD. All gene-specific primers used for the amplification are listed in Appendix A, and *BnaENTH* was used as an internal reference gene [46]. 

### 4.7. Identification of EngA and PsbA Homologs in B. napus

Homologous genes of *EngA* and *PsbA* in *B. napus* were identified, using orthologous and BLAST tools of the BnPIR website [42]. The structural diagram for the *EngA* homologs was mapped using GSDS 2.0 [47].

### 4.8. Protein Subcellular Localization and GUS Staining

The *BnaC6.FtsH1*-coding sequence (using primers C6SL-2F and C6SL-2R) was inserted into the CaMV 35S promoter containing a pM999-GFP vector [45] using single restriction endonucleases (Xba I) and a ClonExpress II One Step Cloning Kit (Vazyme, Nanjing, China). The constructed vector was confirmed correct by sequencing and then introduced into *A. thaliana* (ecotype Col-0) protoplast by PEG-calcium-mediated transfection, then the transfected protoplasts were cultured for 5–24 h [48]. Chloroplast autofluorescence was determined, and the GFP signal was detected using a laser scanning confocal microscope FV1200 (Olympus, Tokyo, Japan) [49] with a green channel, 488 nm excitation wavelength, and a 500–530 nm emission wavelength, respectively.

The promoter containing a 661-bp upstream region of *BnaC6.FtsH1* was amplified (primers C6-P661+Sal1-F and C6-Pro+BamH1-R) and cloned into the pCAMBIA2300 fused with GUS; the resulting plasmid was introduced into *A. thaliana* (ecotype Col-0). *A. thaliana* was transformed using the floral-dip method [50]. Different tissues of *Arabidopsis* T_1_ transgenic lines were stained overnight with X-Gluc solution at 37 °C, and then washed with ethanol as previously described [51]. 

### 4.9. Phylogenetic Analysis

Homologs of *FtsH1*, *FtsH2*, *FtsH5*, and *FtsH8* were searched using the TAIR website (https://www.arabidopsis.org/index.jsp, last accessed date 7 December 2020), the *Brassica* Database [44], and the BnPIR website [42]. Phylogenetic trees were constructed using MEGA X [52]. In detail, protein sequences were aligned using MUSCLE [53], and their evolutionary history was inferred using the maximum likelihood method [54] and a Jones–Taylor–Thornton matrix-based model [55]. In the software interface, nearest-neighbor interchange was used for a heuristic search, while make initial tree automatically (Default-NJ/BioNJ) was used to obtain the initial tree. To assess the reliability of the phylogenetic tree, a bootstrap test [56,57] was performed with 1000 replicates.

## Figures and Tables

**Figure 1 ijms-22-02087-f001:**
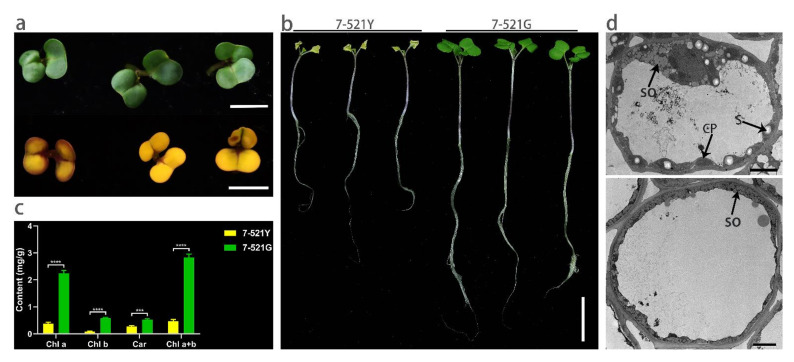
Phenotype characterization of 7-521Y. (**a**) Cotyledon phenotype of 7-521G (up) and 7-521Y (down) in the seedling stage (after 5 days of germination). (**b**) Whole plant phenotype of 7-521G (three on the right) and 7-521Y (three on the left) after germination for 12 days. (**c**) Chlorophyll a, chlorophyll b, carotenoid, and chlorophyll a+b content in 7-521Y and 7-521G. Car/Chl represents the carotenoid/chlorophyll ratio. Asterisks indicate a significant difference: *** *p* < 0.001 and **** *p* < 0.0001 (Student’s *t*-test). (**d**) Transmission electron microscope (TEM) micrographs of the cotyledons from 7-521G (up) and the mutant 7-521Y (down). Black arrows point to different structures in the cell. S—starch grains, CP—chloroplast, and SO—seed oil. Bars: (**a**) 5 mm; (**b**) 2 cm; (**d**) 5 μm.

**Figure 2 ijms-22-02087-f002:**
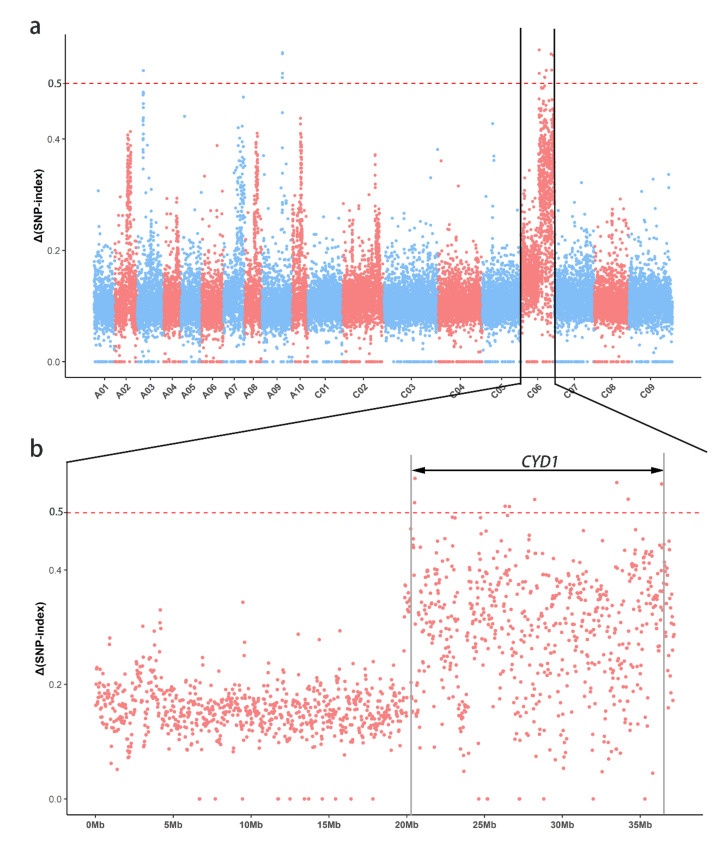
Candidate regions of *CYD1* identified by whole-genome resequencing. (**a**) The x-axis represents the 19 chromosomes of *Brassica napus* arranged by physical position. The y-axis represents the value of Δ (SNP index), which was calculated by subtracting the SNP index of 7-521Y from that of 7-521G. (**b**) Enlarged view of chromosome C06. The candidate gene interval (marked by arrow) was determined using 0.5 as the threshold, and chromosome A03 and chromosome A09 were excluded through marker development.

**Figure 3 ijms-22-02087-f003:**
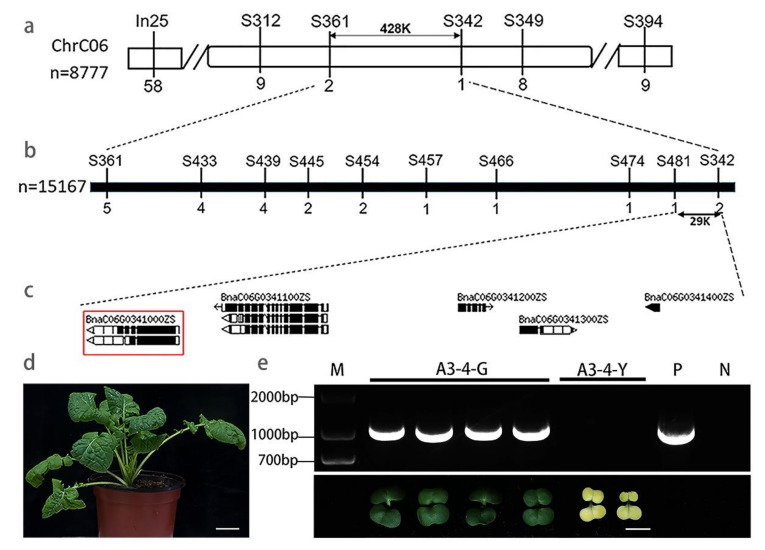
Fine mapping and functional confirmation of *BnaC06.FtsH1* in rapeseed. (**a**) Initial mapping of *CYD*1 using 8777 etiolated-cotyledon individuals from the selfing segregation population of 7-521. The applied markers are marked above the chromosome, and the number of recombinants in this population is marked below. (**b**) Fine mapping of *CYD1* using 15,167 etiolated individuals. The numbers indicate the number of recombinants. (**c**) Annotation in 29K region according to the ZS11 (*Brassica napus* cultivar) reference genome. The red box indicates the candidate genes predicted in ZS11 database. (**d**) Confirmation of the *BnaC06.FtsH1* function using the expression of *BnaC06.FtsH1* in 7-521Y. The phenotype of the T_0_ transgenic plants returned to normal. (**e**) Identification of the T_1_ transgenic seedlings using PCR. A3-4-G, normal individuals from the A3-4 T1 population; A3-4-Y, etiolated individuals from the A3-4 T1 population; N, negative control from 7-521Y; P, positive control from vector plasmid; M, molecular markers. Bars: (**d**) 4 cm; (**e**) 5 mm.

**Figure 4 ijms-22-02087-f004:**
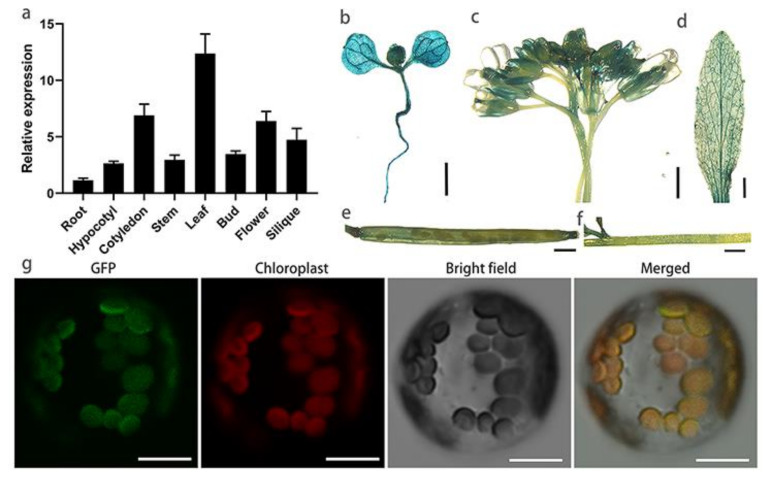
Subcellular localization and expression pattern of *BnaC06.FtsH1*. (**a**) Expression pattern of *BnaC06.FtsH1* detected by qRT-PCR in the hypocotyls, roots, stems, cotyledons, leaves, flower buds, siliques, and flowers of the *Brassica napus* cultivar Westar, using the expression of *BnaENTH* as a reference. Values are expressed as average ± SD (n = 3). (**b**–**f**) GUS staining analysis of *BnaC06.FtsH1*. (**b**) Tender seedling, (**c**) bud, (**d**) leaf, (**e**) silique, and (**f**) stem. (**g**) Subcellular localization of BnaC06.FtsH1 in *A. thaliana* protoplasts. From left to right: green fluorescence of BnaC06.FtsH1; chloroplast spontaneous red fluorescence; bright-field image; merged image of the first three images. Bars: (**b**–**f**) 1 mm; (**g**) 12 μm.

**Figure 5 ijms-22-02087-f005:**
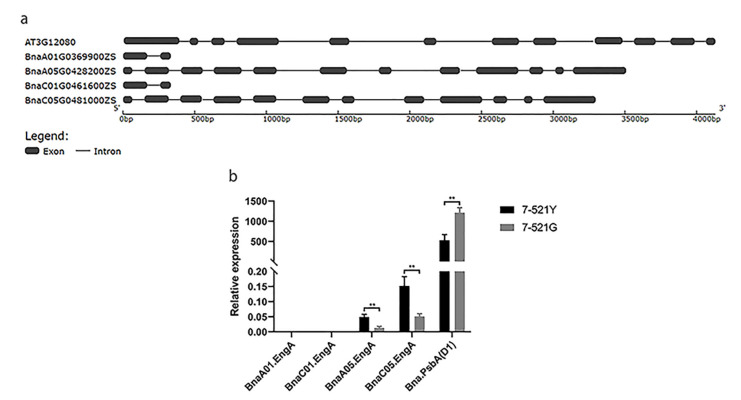
Expression levels of *EngA* and *PsbA* determined by qRT-PCR in *Brassica napus*. (**a**) Homologous gene structure analysis of *EngA* in *B. napus*. (**b**) Relative expression of *EngA* and *PsbA* homologous genes in *B. napus*. ** Denotes significant differences, *p* < 0.001, Student’s *t*-test.

**Figure 6 ijms-22-02087-f006:**
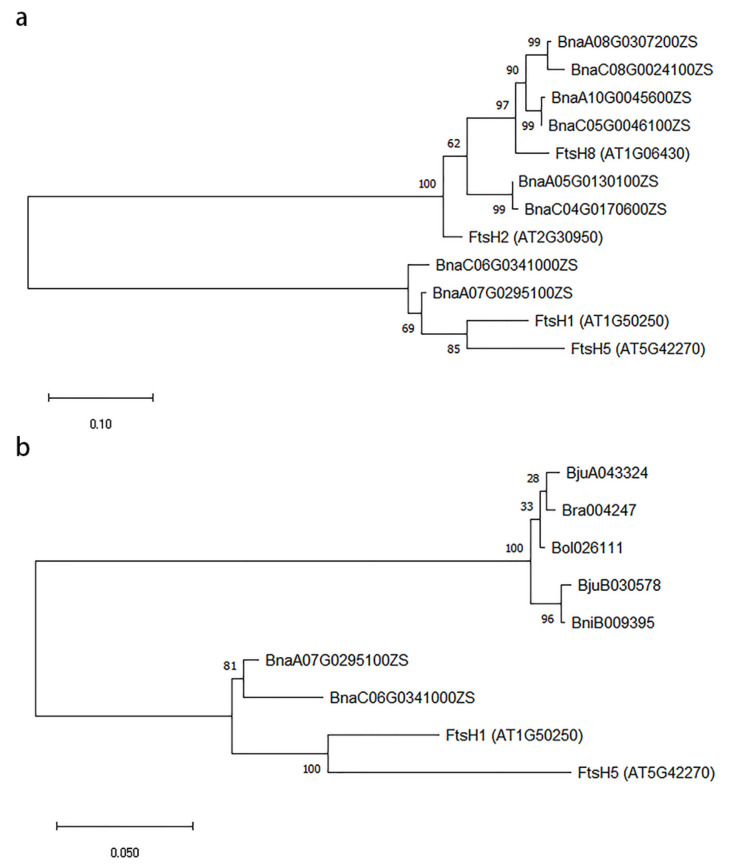
Construction of two maximum likelihood phylogenetic trees. (**a**) Phylogenetic analysis of filamentation temperature sensitive protein H 5 (*FtsH5*), *FtsH1*, *FtsH8*, *FtsH2*, and their 8 closest homologs in *Brassica napus*. (**b**) Phylogenetic analysis of type A (*FtsH1* and *FtsH5*) and their closest homologs in *B. rape*, *B. nigra*, *B. juncea*, *B. oleracea*, and *B. napus*. The tree is drawn to scale, with branch lengths measured as the number of substitutions per site. The length of these branches refers to the rate of sequence variation. Numbers represent the reliability calculated by the bootstrap test with 1000 replicates.

**Table 1 ijms-22-02087-t001:** Prediction and annotation of candidate genes within the mapping region.

Gene of *B. napus*	Chromosome Position	Orthologous Gene of *A. thaliana*	Annotation
BnaC06G0341000ZS	44,665,913–44,669,710	AT1G50250	FtsH protease 1
BnaC06G0341100ZS	44,671,450–44,675,781	AT1G67720	Leucine-rich repeat protein kinase family protein
BnaC06G0341200ZS	44,681,149–44,682,236	AT4G01935	insulin-induced protein
BnaC06G0341300ZS	44,683,655–44,685,971	AT1G67730	beta-ketoacyl reductase 1
BnaC06G0341400ZS	44,688,777–44,689,358	AT1G67740	PsbY precursor (psbY) mRNA

## Data Availability

All data generated or analyzed during this study are included in this published article and its Appendix A.

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
