# Peer review of "Fine Mapping and Identification of BnaC06.FtsH1, a Lethal Gene That Regulates the PSII Repair Cycle in Brassica napus"

_ijms, 2021, doi:10.3390/ijms22042087_

Round 1

Reviewer 1 Report

Dear Editor/Author,

The construction of the photosynthetic apparatus is closely related to the light conditions in the environment, and the adaptation of plants to different and changing light intensities attracts behind them changes in the structure and functioning of photosynthetic complexes.The authors of the paper: "Fine mapping and identification of BnaC06.FtsH1, a lethal gene that regulates the PSII repair cycle in Brassica napus" provided new information about the lethal mutants which information can be used to increase PSII repair performance.However, I have stipulation about several fragments of manuscript. I recommended this paper to publication in Agriculture, as “ Reconsider after major revision (control missing in some experiments)”. However, I have stipulation about several fragments of manuscript. I recommended this paper to publication in International Journal of Molecular Sciences as “ Accept after minor revision (corrections to minor methodological errors and text editing)”.

1- In Abstract- results and achievements are not shown.

2- In introduction - there is no description of the purpose of the analysis and the benefits resulting from it (mainly for breeders)

3- In discussion - at the end of the discussion there is no list of the most important results obtained in the work.

Reviewer 2 Report

Dear Authors,

I have reviewed the manuscript "Fine mapping and identification of BnaC06.FtsH1, a lethal gene that regulates the PSII repair cycle in Brassica napus" that describes the identification of a FtsH1 that seems to encode an FtsH1 hydrolase that degrades damaged PSII D1. The manuscript is well written. the introduction is complete as the results and the methods. In my opinion, the discussion should be implemented as it does not discuss at all about the second candidate CYD2. 

there are some minor comments to address:

Line 28: change period with semicolon.

Line 37: check the double commas

Line 59: add a space after the period

Line 61: add a space after the period

Line 81: replace “mutants” with “mutant” at the beginning of line

Line 146: check “Individuals”

Line 163 (Table1 legend): check “regio”

Line 241: Check the “we combined…”

Line245: check “transformed”

My conclusion is to resubmit the manuscript after the major revision. I think that some discussion about the CYD2 candidate (of which you showed some data) is mandatory. 

Kind regards

Reviewer 3 Report

The manuscript entitled “Fine mapping and identification of BnaC06.FtsH1, a lethal gene that regulates the PSII repair cycle in Brassica napus” is quite interesting and could have a scientific impact on plant science, however before publication some changes should be made.

In the Introduction part of the manuscript, the authors describe the CRISPER-Cas9 system but there is no connection with the obtained results. Please explain.

The authors should provide a better resolution of Fig. 2

In Table 1. Latin names should be in italic. Please correct.

In the Materials and methods part of the manuscript please provide a model of used instruments together with the vendors, city, and country of origin.

Section 4.5. Detail method concerning transformation should be added. There are two transformations, in bacteria and plant. Please describe the protocols for transformations, is chemical or electroporation mean of transformation are used, which were the condition, etc.

Line 351 Please add space between value and unit.

There are some inconsistencies in the reference style. Please carefully check and correct according to the journal instructions

Reviewer 4 Report

REVIEW OF THE ARTICLE BY LAI XU ET AL. ENTITLED 'FINE MAPPING AND IDENTIFICATION OF BNAC06.FTSH1, A LETHAL GENE THAT REGULATES THE PSII REPAIR CYCLE IN BRASSICA NAPUS' (ijms-1093260)

Lai Xu et al. created the lethal mutant 7-521Y of Brassica napus with etiolated cotyledons. They reseqenced genome of the mutant, mapped the cyd1 gene responsible for the ethiolated phenotype. They found that BnaC06.FtsH1 encoding the FtsH1 hydrolase is the target gene involved in the damaged PSII reaction centers changeover, indicate expression patterns of the gene, verified its function and subcellular localization of the product, perform phylogenetic analysis of the gene (or its product). 

The data are new and interesting. Article is in the scope of the Journal. Introduction provides all necessary background of the study as well as the goal. Results, most of the results are clearly reported, all methods used in the work are listed in the corresponding subsection, but their description should be improved, see specific comments). It would be nice to see more detailed discussion of valuable results obtained by the Authors.

Collectively,  text is well-written and interesting. Data are valuable. The article can be published after a revision in accordance with specific comments(see below). My main criticisms are addressed mainly to MAterials and Methods and Discussion.

LIST OF SPECIFIC COMMENTS

In accordance with the Journal's rules, 'Fig.' should be written in full - 'Figure'.

l. 37. If You use latine name for Arabidopsis, it is better to use it for wheat, cotton and tobacco as well.

l. 84-85. Please, describe results of TEM more detail. "chloroplast of the mutant was abnormal and that starch synthesis was blocked" - why? How was this tatement confirmed by TEM images. As for me, TEM images substandard: separate thylacoid memranes and other intracellular structures are not distyinguished. Images are not focused. If possible, please, replace them by high quality images. Indicate more structures on the figure, e.g. vacuole.

l. 87. Part of the text is bold.

Fig. 1. 'starch' should be 'starch grains'.

l. 104. What was the average coverage of the reference genome?

l. 130. "was highly orthologous to" - please, describe it clearly, why it is not paralog.

l. 136. "full-length genome" - did You mean 'complete genome'?

Table 1. A. thaliana shold be italicized.

l. 139-148. Although the Authors performed photosynthetic pigment assay, these data are not clearly presented, they only discuss the results in terms 'green' and 'yellow' phenotype. It is better to confirm it in the text and figures by pigmrnt content data, e.g. the Car/Chl (mg/mg or mol/mol) ratio. This index with the Chla/Chlb ratio is a sign of stress. Therefore it is possible to conclude about physiological state of plants.

l. 173-176. It is Methods.

l. 190. Please, indicate in Methods alignment method with corresponding reference. BLAST?

l. 221-229. The content of the last subsection of the Results is highly speculative and not confirmed by the data. Title of the subsection should be about phylogenetic analysis.  Dendrograms prewsented are not a confirmaqtion of the facts listed in this subsection. Moreover it is not results. It is discussion. Please, try to be more descriptive in this part of the article. describe dendrograms and phylogenetic relationship of proteins.

Fig. 6 is not unified. There is a cladogram (on Fig. 6a) and phylogram (on Fig. 6b). As I see from the figure, either the Authors constructed trees by different algorhytms or ignored phylogenetic distances on Fig. 5a. Please perform phylogenetic analysis by the same algorhytms and provide two phylograms. In the capture, please, describe scale bars and numbers above nodes.

Please, involve an outgop into the phylogenetic analysis.

l. 237-249. It is summary of results, not discussion. It is excessive and should be eliminated.

In general, Discussion is poore. The Authors obtain new valuable data and presented them, but not discussed it in detai. I suggest, in the Discussion section, provide point-to-point discussion of all obtained results with cxorresponding references.

l. 291. Popint is missing.

l. 292. Since the Authors determine not only chlorophyll but also total carotenoid concentration, it is better to change the title of the subsection into eg "Pigment assay". Was it necessury to grid plant tissues with liquid nitrogen for effective pigment extraction? Please, comment.

l. 294. Do You mean actually 96% ethanol. As a rule labour ethanol is 96%. Indicate type of the ratio: was it either volumetric or mass.

l. 298-299. Please, revise equations. Now they are confusing: what is the units for pigment concentrations in the extracts, what is 'weight' and what are its units, what do all symbols in the equation mean (Chl a, Chl b, Car, A...)? it should be described.

l. 298-299. If the equations are from a published source, the reference should be added.

l. 293-299. Please, indicate the type of spectrophotometer for absorgbance determination, type of couvette and optical pathlength. Write also that, pigment content was determined spectrophotometrically. 

l. 302. PH should be pH.

l. 302. Was the buffer ether Na-phosphate or K-phosphate?

l. 308. Please, indicate, whether it was one-end or pair-end Illumina sequencing.

l. 305-316. Some experimental details should be described more detai: did the Authors perform trimming and quality analysis of reads datasets by eg FastQC?, how were Illumina libraries were prepared?, indicate reagents for Illumina sequencing, did You perform assesment of gDNA concentration and quality before the sequencing? Which genome browser was used?

l. 324. Please, provide the reference for the medium.

l. 320, l.342. Please, describe the procedure of clonning and vector obtaining.

l. 331. Although it is common, to be completely justified, please, indicate, how did You calculate 'relative expression'.

l. 343. Please, provide line/strain of the A. taliana. Please, provide the procedure or reference for protoplast obtaining.

l. 345-346. Please, describe more detail evaluation of the samples by confocal microscopy: type of the objective, excitation light, detection channel, etc.

l. 352-355. I am disappointed a bit that phylogenertic analysis generally has not been described in Methods at all. It was described only that the analysis was conduced in the MEGA X. It is OK, but also it is necessury to indicate was the analysis conduced based on nucleotide sequences or sequences of encoded proteins; the algorithm of multiple alignment, procedure of evolution model selection, algorhythm of tree reconstrruction mast be provided with corresponding references. If the Authors used the maximim likekihood algorhythm, they should also provide the method for initial tree obtaining and heuristic search. As I see from results, the Authors also performed bootstrap procedure for assessment of tree topology robustness. It should be indicated in the mehthods with corresponding reference to the bootstrap procedure. Indicate the number of bootstrap replicates.

Obtained sequences of FtsH5, FtsH1, FtsH8, FtsH2, Ftsh1 and Ftsh5 obtained in the work should be deposited into the NCBI GenBank.

Round 2

Reviewer 2 Report

Dear Authors,

thank you for providing an improved version of the manuscript. In my opinion the manuscript is now suotable for publication with just this comments to be checked: 

Line 14_ Check “cotyleifgdons”. I do know how but in the previous revision it was correctly written

Lines 113 and 114: do not use the space between the number and the x when indicating coverage depth

Line 217: A period is missing at the end of the sentence.

Line 283. Check “we” for capital letter

Line 367: change “pst I” with “Pst I”

Best regards

Reviewer 3 Report

Authors made all changes according to my suggestions, thus the paper should be accept in the present form.

Author Response

Thanks for your kind advice and detailed suggestions.

Reviewer 4 Report

REVIEW OF THE ARTICLE BY LAI XU ET AL. ENTITLED 'FINE MAPPING AND IDENTIFICATION OF BNAC06.FTSH1, A LETHAL GENE THAT REGULATES THE PSII REPAIR CYCLE IN BRASSICA NAPUS' (ijms-1093260) - REVISED VERSION

The Authors signifficantly improved their manuscript. Most of my suggestions were taken into account. However there are still some important issues. Particularly, the texst still not completely structurized: some part of results are actually methods, if the Authors decided to present Results and Discussion separately, it is necessury also move some sentences to the Discussion. Results should not contain references. Discussion of obtained results is still not complete. Please, see lthe list of my specific comments for revision.

SPECIFIC COMMENTS

l. 91-93: these structures should be indicated on figures.

l. 118: it is methods.

Figure1: the units of pigment content and Car/Chl ratio are not the same (mg/g of leaf and mg/mg, respectively), therefore, it is incorrect to present them to the same scale.

l. 142: 'blast' should be 'BLAST'. Moreover, it is methods.

l. 132-152.Please, move reverences to on-lone resources and software to Methods.

l. 204-205. It is methods. 'Blast' should be 'BLAST'.

l. 203-204. It is not results. It is ether Discussion or Introduction.

l. 208. It is Methods (information about GSDS).

l. 231-234. It is Methods.

l. 263. 'cndition' - typing error.

-Discussion is still not complete. Please, discuss also ultrastructural and pigment data. Particularly, it is important, that PSII pigment-protein complexes are responsible for thylakoid stacking. Therefore absence of assembled granae may be associated fith PSII reaction centers defficient. The same is true about Car/Chl ratio: PSII is characterized strict stoicheometry between photosynthetic pigments: thus, its shift is an indicator of functional PSII dissasembly.

l. 354: it is better to indicate 'software [36], respectively'.

l. 410-418: the description of phylogenetic analysis has been signifficantly improved, but there are some issues. Which information criterion did the Authors use for model selection (either Bayesian or Akaike)? They indicated, that the trees were reconstructed by the ML method, therefore in the figure capture 'bootstrap negbor-join' should be replaced to 'Maximum Likelihood'. Isn't it?

Round 3

Reviewer 4 Report

I agree with all the changes in the manuscript. Now the paper is acceptable.